# SMARCB1-Deficient Cancers: Novel Molecular Insights and Therapeutic Vulnerabilities

**DOI:** 10.3390/cancers14153645

**Published:** 2022-07-27

**Authors:** Garrett W. Cooper, Andrew L. Hong

**Affiliations:** 1Department of Pediatrics, Emory University School of Medicine, Atlanta, GA 30322, USA; garrett.cooper@emory.edu; 2Aflac Cancer and Blood Disorders Center, Children’s Healthcare of Atlanta, Atlanta, GA 30322, USA; 3Winship Cancer Institute, Emory University School of Medicine, Atlanta, GA 30322, USA

**Keywords:** SMARCB1-deficient cancer, structure, therapeutics, rhabdoid tumor, epigenetics, chromatin

## Abstract

**Simple Summary:**

Loss of SMARCB1 has been identified as the sole mutation in a number of rare pediatric and adult cancers, most of which have a poor prognosis despite intensive therapies including surgery, radiation, and chemotherapy. Thus, a more robust understanding of the mechanisms driving this set of cancers is vital to improving patient treatment and outcomes. This review outlines recent advances made in our understanding of the function of SMARCB1 and how these advances have been used to discover putative therapeutic vulnerabilities.

**Abstract:**

SMARCB1 is a critical component of the BAF complex that is responsible for global chromatin remodeling. Loss of SMARCB1 has been implicated in the initiation of cancers such as malignant rhabdoid tumor (MRT), atypical teratoid rhabdoid tumor (ATRT), and, more recently, renal medullary carcinoma (RMC). These SMARCB1-deficient tumors have remarkably stable genomes, offering unique insights into the epigenetic mechanisms in cancer biology. Given the lack of druggable targets and the high mortality associated with SMARCB1-deficient tumors, a significant research effort has been directed toward understanding the mechanisms of tumor transformation and proliferation. Accumulating evidence suggests that tumorigenicity arises from aberrant enhancer and promoter regulation followed by dysfunctional transcriptional control. In this review, we outline key mechanisms by which loss of SMARCB1 may lead to tumor formation and cover how these mechanisms have been used for the design of targeted therapy.

## 1. Introduction

All eukaryotic cells arrange their genomic DNA into a highly organized and compact structure known as chromatin. Spatial compaction occurs when ~147 base pairs of DNA wrap around a histone protein octamer to form the basic subunit of chromatin, known as the nucleosome. These nucleosomes become further compacted as they aggregate into higher-order structures such as chromosomes. For gene expression to properly occur, transcriptional activators need to physically bind to regulatory DNA elements such as enhancers and promoters. To promote accessibility, nucleosomes need to be repositioned so that the underlying regulatory DNA sequences are unwound. Nucleosome repositioning can be mediated by a variety of chromatin-remodeling complexes such as BAF, CHD, INO80, and ISI. The BAF (BRG1/BRM-associated factor) complex has been found to play a critical tumor-suppressive role in humans and is mutated in up to 25% of cancers. This review is focused on one of these BAF subunits, SMARCB1, which has been identified as the sole recurrent genetic alteration present in a variety of cancers known as SMARCB1-deficient cancers.

The protein SMARCB1 can be referred to by a variety of names originating from the context of its discovery: SNF5, INI1, and BAF47. This review uses the Human Genome Organization-derived name of SMARCB1, which abbreviates the full gene name of *SWI/SNF Related, Matrix Associated, Actin Dependent Regulator of Chromatin, Subfamily B, Member 1*. SMARCB1 is a highly conserved core subunit of the mammalian ATP-dependent BAF chromatin remodeling complex, a key regulator of nucleosome positioning and gene expression. *SMARCB1* was initially identified in a screen of *Saccharomyces cerevisiae* as a gene required for sucrose fermentation in 1984, subsequently named Snf5 (sucrose-nonfermenting 5) in 1990 [1,2]. The human ortholog, originally termed INI1 (integrase interactor 1), was subsequently shown in 1994 to bind and stimulate HIV-1 integrase in vitro [3]. Fourteen years after its initial discovery in *S. cerevisiae*, *SMARCB1* was characterized in humans as a bona fide tumor suppressor gene biallelically inactivated in particularly rare and lethal early childhood cancers known as malignant rhabdoid tumor of the kidney (MRT) and atypical teratoid rhabdoid tumor (ATRT) [4].

Modern advances in diagnostic techniques have identified SMARCB1 loss in numerous other cancers broadly referred to as SMARCB1-deficient cancers [5]. These tumors have surprisingly stable genomes, with *SMARCB1* often being the only genetic alteration present. This genomic stability contrasts with the idea that cancer is the accumulation of DNA mutations that affect the function of numerous genes [6], instead revealing that global changes in epigenetic regulation can potently transform diverse cell types into highly lethal and malignant tumors [7]. While the role of SMARCB1 within the BAF complex is not entirely understood, it appears to facilitate BAF complex binding and promote its remodeling activity at key regulatory regions such as enhancers and promoters. Upon loss of SMARCB1, many of these regulatory regions become inactive, which leads to genome-wide transcriptional deregulation.

There is still much to understand about the downstream effects of SMARCB1 loss, but recent advancements have furthered our understanding of the molecular mechanisms of oncogenesis. For example, SMARCB1 has been identified as a critical regulator of genome-wide enhancer activation but does not appear to affect the activity of super-enhancers. The identification of novel SMARCB1 interactions has expanded our understanding of how it functions as a tumor suppressor. Advancements in our ability to perform large-scale screens have led to the discovery of various therapeutic vulnerabilities that were previously unknown. This review covers major advancements in each of these areas and discusses gaps in our knowledge that remain to be answered.

## 2. SMARCB1-Deficient Cancers

The first indication that the loss of a tumor suppressor may be involved in atypical teratoid-rhabdoid tumors (ATRTs) came from cytogenetic characterizations of these tumors in 1990. Chromosomal analysis suggested that the sole cause of these rhabdoid tumors was the loss of genes on chromosome 22, and subsequent studies showed more specific alterations in chromosome band 22q11.2 [8,9]. It was later shown in 1998 that biallelic alterations to SMARCB1 were present in the vast majority of MRT cases (90%), consistent with the recessive “two-hit” tumor suppressor model of oncogenesis [4]. Since then, widespread sequencing efforts and the use of immunohistochemistry (IHC) have identified numerous other cancers with the depletion or loss of SMARCB1 [10,11]. Current estimates indicate that 1.4% of all cancers contain SMARCB1 alterations (1152 out of 84,646 queried samples found on AACR Project GENIE Cohort v11.0) [12]. However, a recent meta-analysis examining 10,849 patients from 15 studies found that 5% of cases had alterations in SMARCB1 [13].

### 2.1. Types of SMARCB1-Deficient Cancers

SMARCB1-deficient cancers are characterized by the biallelic loss of function in both *SMARCB1* alleles. The prototypical SMARCB1-deficient cancer is MRT, but numerous other tumor types with the complete loss of SMARCB1 have been described such as renal medullary carcinoma (RMC), epithelioid sarcoma (ES), and pediatric poorly differentiated chordoma, almost all of which have a poor prognosis [11,14,15,16,17]. Other cancers with disrupted SMARCB1 function include synovial sarcoma (SS), myoepithelial carcinomas, and sinonasal carcinomas [18,19,20,21,22,23,24]. Interestingly, schwannomatosis and cribriform neuroepithelial tumor have also been found to have alterations in SMARCB1, but overall outcomes remain high [16,25].

### 2.2. Initiation of SMARCB1-Deficient Tumors

SMARCB1 was first confirmed as a potent in vivo tumor suppressor when a conditional gene knockout of *Smarcb1* led to fully penetrant cancer formation in adult mice, resulting in CD8+ mature lymphoma [26]. While this model demonstrates the potent tumor-suppressive function of SMARCB1, the resulting lymphomas do not offer a faithful model for most human SMARCB1-deficient cancers. The unexpected development of lymphomas may have been due to reduced knockout efficiency in the adult brain or kidney and the late developmental stage at which *Smarcb1* deletion was induced. Later studies were able to faithfully reproduce ATRT-like tumors by inducing SMARCB1 deletion in earlier embryonic developmental time points (between E6 and E10) with Cre expression driven by the ubiquitous *Rosa26* promoter [27]. This age-specific development of ATRT suggests that transient populations of neural progenitor cells are capable of transformation upon *Smarcb1* loss.

In fact, recent studies have proposed that rhabdoid tumors arise from a small population of neural crest cells that lose SMARCB1 during development [28]. This model suggests that loss of SMARCB1 blocks the ability of neural crest cells to differentiate into mesenchyme, locking them in a partially differentiated state. This hypothesis is further supported by the observation that the re-expression of SMARCB1 in rhabdoid tumor cells leads to the upregulation of genes associated with the epithelial to mesenchymal transition [29]. This proposed mechanism raises a critical question: could pushing these tumors toward a more mesenchymal cell state, akin to SMARCB1 re-expression, be a therapeutic target? While there are no current methods to restore SMARCB1 function in tumors, this mechanism offers a way to mimic the effects of SMARCB1 reconstitution. Further studies are needed to validate these mechanisms, but current studies offer promising new insights that may be used for future therapeutics [30].

Familial cases of rhabdoid tumor can occur when an individual has a condition known as rhabdoid tumor predisposition syndrome (RTPS). In RTPS, a germline variant arises in either *SMARCB1* (known as RTPS1) or the ATPase subunit found in the BAF complex, *SMARCA4* (known as RTPS2) [31,32]. Having a germline mutation in one allele of either *SMARCB1* or *SMARCA4* increases the risk of developing rhabdoid tumors in children and schwannomas in adults because only one functional copy is present in each cell.

While these findings confirm the developmental origin of rhabdoid tumors, it fails to elucidate how SMARCB1-deficient cancers arise in adults. Both ES and RMC diagnosis occur in adolescents and young adults [33,34]. The origins of these two cancer types are much less understood. RMC arises in the kidney and is known to be associated with the sickle cell trait, although the mechanism by which this predisposition promotes tumor formation is unknown [35]. Even less is known about the initiation of ES, except that 90% of cases have loss of SMARCB1 and it typically arises in extremities [36]. Further work is needed to understand how loss of SMARCB1 drives tumor formation in ES and RMC as compared to rhabdoid tumors.

### 2.3. Unique Role of SMARCB1 in Synovial Sarcoma

Synovial sarcoma (SS) is a rare cancer of mesenchymal origin that frequently arises in soft tissues of the extremities [37]. SS is driven by a recurrent chromosomal translocation, t(X; 18)(p11.2; q11.2), that fuses the SS18 gene on chromosome 18 with a related SSX gene located on the X chromosome, forming the *SS18-SSX* fusion protein [38,39,40]. Similar to how SMARCB1 inactivation is the sole driver of MRT, this translocation is present in 95% of SS cases [41]. Interestingly, synovial sarcoma is also defined by the loss of SMARCB1 protein levels when observed by immunohistochemistry [18]. However, synovial sarcoma samples with reduced SMARCB1 protein levels had high levels of *SMARCB1* mRNA [18]. This suggests a post-transcriptional role for the degradation of SMARCB1 in the setting of synovial sarcoma.

A proposed mechanism for this SMARCB1 depletion in SS is that the SS18-SSX fusion protein outcompetes SMARCB1 binding in BAF-family complexes, leading to SMARCB1 ejection from BAF and subsequent degradation [42]. Under this model, known as the SMARCB1 ejection model, BAF complexes lacking SMARCB1 are aberrantly targeted to proliferative genes, such as *Sox2*, driving the overexpression and proliferation of cells. Subsequent studies have suggested a more nuanced understanding of SMARCB1 in SS. In this newer model, SMARCB1 is still able to associate with SS18-SSX containing BAF complexes [23]. Instead, these SS18-SSX fusion BAF complexes bound to SMARCB1 are targeted for whole-complex proteasomal degradation through an unknown mechanism, rendering the cells BAF-deficient. In response, cells upregulate the formation of another BAF complex lacking SMARCB1, known as GBAF (GLTSCR1/like-containing BAF), as a survival mechanism. This recently proposed BAF deficiency model agrees with the GBAF-specific BRD9 dependency found in SS as well as other SMARCB1-deficient tumors. This BRD9 dependency is further discussed in Section 7.2 [43,44].

## 3. Structure of SMARCB1

SMARCB1 is a 47kDa nuclear protein with a length of 385 amino acids. Structurally, it has four distinct domains: an N-terminal Winged Helix DNA binding domain (WHD), followed by two highly conserved repeat domains referred to as Repeat (RPT) 1 and 2, and a C-terminal coiled-coil domain (CTD). There have been many proposed biological interactions with each of these regions, and in this section, we review recent findings and potential questions that remain (Figure 1).

### 3.1. Winged Helix Domain

Germline missense mutations and in-frame deletions within the WHD of SMARCB1 have been linked to schwannomatosis, a condition predisposing patients to forming benign tumors that develop in the CNS called schwannomas [45,46,47,48,49]. These mutations contrast with the large deletions or truncations found in many other SMARCB1-deficient cancers. The WHD of SMARCB1 shows structural similarity to the winged helix domain found in many DNA binding proteins, particularly the well-characterized MBP1 family of yeast cell cycle regulators [50,51,52]. In fact, NMR studies suggest that the SMARCB1 WHD alone was sufficient to bind to dsDNA [51]. Despite these findings, structural studies indicate that the WHD of SMARCB1 is deeply buried within the BAF complex far away from nucleosomal DNA [53,54]. Given these seemingly conflicting pieces of data, it remains unclear how the WHD of SMARCB1 functions within the BAF complex and why germline mutations in the WHD predispose patients to tumor formation.

### 3.2. Tandem Repeat (RPT) Domains

SMARCB1 contains two highly conserved ~60-amino-acid imperfect repeat regions referred to as RPT1 and RPT2. RPT1 consists of a two-stranded antiparallel β-sheet followed by two α-helices [55]. Both repeat domains appear to be critical for various protein-protein interactions including cellular machinery and viral proteins [56,57,58,59]. Perhaps most notably, RPT1 has been shown to be necessary for MYC and HIV-1 integrase (IN) binding to SMARCB1 [3,56,60] and RPT2 has been shown to interact with XPO1 via a nuclear export signal [61]. These interactions are discussed further below.

The RPT2 domain is a highly conserved region composed of a three-stranded antiparallel β-sheet and two α-helices. A recent study utilizing both cryo-EM structures alongside immunoprecipitation assays suggests that RPT2 is required for DPF2 association with the BAF complex [54]. They provide robust molecular data demonstrating that mutant SMARCB1 constructs lacking RPT2 are unable to associate with DPF2. Conflicting cryo-EM studies, notably with higher resolution, postulate that both RPT1 and RPT2 interact with SMARCC2 as opposed to DPF2, perhaps as a mechanism to stabilize core BAF complex formation [53,55]. These differing hypotheses raise the possibility that the RPT2 region of SMARCB1 has varied context-dependent interactions, although further study is needed to fully characterize its interactions. A nuclear export signal within the RPT2 domain has also been identified, which is discussed in further detail in Section 5.3.

### 3.3. C-terminal Coiled-Coil Domain (CTD)

The CTD of SMARCB1 is composed of a dense region of basic amino acids that has been shown to physically interact with the nucleosome acidic patch opposite of SMARCA4 [62]. The CTD has been shown to have mutations that are associated with Coffin-Siris-syndrome, a rare intellectual disability [63,64,65]. Cancer-associated mutations within the CTD have also been found in meningioma, adenocarcinoma, and schwannoma, particularly R374Q and R377H, but low-frequency cancer-associated mutations are found throughout *SMARCB1* [66,67,68]. Due to the relatively low number of currently sequenced patient tumor samples with SMARCB1 mutations, it is difficult to identify the pathogenic variants due to low signal. To circumvent this difficulty, high-throughput methods can be used to identify pathogenic SMARCB1 variants in vitro [69].

### 3.4. Conservation of SMARCB1

SMARCB1 is conserved in nearly all eukaryotic species [70]. The winged helix domain appears dispensable within the plant species, *A. thaliana*, while the other three domains, RPT1, RPT2, and CCD, have numerous conserved regions throughout (Figure 2).

## 4. SMARCB1 and the BAF Complex

The BAF complex is one of four mammalian ATP-dependent chromatin remodelers, the others being INO80/SWR1, CDH, and ISI [71]. The BAF complex is composed of at least 15 subunits and is primarily responsible for regulating gene expression and development by positioning nucleosomes at gene regulatory regions [72]. SMARCB1 is a critical subunit of the BAF complex, particularly in complexes that localize to promoters and enhancers. The two BAF subunits SMARCA4 and SMARCB1 directly bind opposing sides of the nucleosome in a mechanism resembling a clamp [62]. The ATPase subunit SMARCA4 then moves the DNA along the nucleosome as SMARCB1 holds the BAF complex tightly to the nucleosome [62,73,74]. Upon loss of SMARCB1, the BAF complex is unable to recognize and bind its target regions, leading to widespread transcriptional deregulation.

### 4.1. Three Different BAF Sub-Complexes

Within the BAF complex family, there are three known subcomplexes: canonical BAF (cBAF) [75,76], polybromo-associated BAF (PBAF) [77,78,79], and the newly characterized GLTSCR1/like-containing BAF (GBAF or ncBAF) [44,80,81,82]. All three of the BAF complex members have distinct functions determined by the incorporation of complex specific subunits [83]. Notably, SMARCB1 is present in cBAF and PBAF complexes while being absent from GBAF complexes [44,81]. SMARCB1 acts as an anchor that binds to the nucleosome acidic patch via a highly basic C-terminal alpha helix, while the ATPase subunit SMARCA2/4 binds to the opposing side of the nucleosome [53,62,84]. With the nucleosome held on both sides, the ATPase subunit is able to slide DNA along the nucleosome. Mutations in the highly basic SMARCB1 C-terminal alpha helix disrupt nucleosome binding and reduces remodeling efficiency [62]. Interestingly, these C-terminal mutations have little effect on global BAF localization, suggesting that this particular interaction is not critical for BAF targeting.

### 4.2. Transcriptional Regulation by SMARCB1

While much prior work has generally implicated BAF complexes in the regulation of promoters [85] and lineage-specific enhancers [86,87,88,89,90], emerging evidence points to distinct localization patterns for each subcomplex. cBAF complexes predominantly localize to active distal enhancers regions [29,91,92], while PBAF complexes are enriched at active proximal-promoters regions [44]. GBAF has been shown to predominantly localize at CTCF motifs and promoters, suggesting a unique regulatory role in CTCF-mediated 3D chromatin architecture [44,82,93,94,95,96].

The cBAF and PBAF-specific subunit SMARCB1 is critical for genome-wide BAF stability on chromatin and on enhancers (Figure 3) [29,91,92]. The reintroduction of SMARCB1 into MRT cell lines substantially increases cBAF localization at distal enhancer sites [29,92], and promotes active enhancer marks such as H3K27ac and H3K4me1 [91]. By promoting these activating marks, enhancers more readily promote the gene expression of target genes by facilitating formation of the preinitiation complex [97,98,99,100]. Similarly, PBAF is preferentially recruited to bivalent promoters upon SMARCB1 expression and promotes the transcription of target genes. BAF is able to promote the expression of transcriptionally poised genes in a matter of minutes by removing the polycomb repressive complex (PRC2) and its repressive H3K27me3 mark from both promoters and enhancers [101,102]. The rapid BAF-dependent activation and repression of bivalent promoters and enhancers allows for an exquisite temporal regulation of gene expression.

Loss of SMARCB1 causes the widespread loss of BAF localization, which leads to unchecked PRC2-mediated transcriptional repression at enhancers and promoters [103,104]. However, there appears to be residual BAF complexes lacking SMARCB1 that are essential to the survival of SMARCB1-deficient lines [105]. Interestingly, these residual BAF complexes are able to maintain active chromatin organization even at regions also bound by EZH2 [106]. Thus, while loss of SMARCB1 generally leads to widespread transcriptional repression, there appears to be a critical subset of genes driven by residual BAF complex activity that is required for cancer progression. This finding raises two potential possibilities. Either the residual BAF remodeling activity is solely carried out by GBAF, or the cBAF and PBAF complexes lacking SMARCB1 can achieve reduced remodeling activity. Nevertheless, it remains generally unclear how BAF is able to achieve remodeling activity while lacking SMARCB1.

### 4.3. SMARCB1 Regulation of Super Enhancers

The role SMARCB1 plays in the regulation of cell lineage-specific super-enhancers is much less well defined. Typical enhancers can range anywhere from 1 to 4 kilobases and typically only regulate a small set of genes [107]. In comparison, super enhancers span large genomic regions between 10 and 60 kilobases and regulate cell identity transcriptional states through extraordinarily high levels of transcription factors and H3K27ac that synergistically combine to regulate gene activity [108,109]. One group reported that SMARCB1 has largely no impact on the accessibility of super-enhancers [92], while others have reported that SMARCB1 re-expression promotes active marks on super-enhancers [29]. A recent study reported that SMARCB1 promotes repressive chromatin at super-enhancers in human embryonic stem cells (hESCs), while still acting as a transcriptional activator at enhancers [110]. This finding raises the nuanced possibility that SMARCB1 acts as a transcriptional repressor only in the setting of hESC super-enhancers. Still though, further work is needed to characterize the role SMARCB1 has in regulating super-enhancers.

### 4.4. SMARCB1-Dependent BAF Complex Stability

The BAF complex stability in the presence or absence of SMARCB1 has yet to be completely resolved. Some studies suggest that loss of SMARCB1 leads to the dissociation of complex subunits [92,111,112], while others suggest that loss of SMARCB1 leaves complex assembly relatively unaltered [29,83,101,105,113,114]. These conflicting results are most likely due to variable cellular fractionation procedures, but accumulating evidence points to residual BAF complex stability without the presence of SMARCB1. Interestingly, even in studies that suggest SMARCB1 does not play a major role in BAF complex stability, DPF2 seems to have a highly SMARCB1-dependent association with BAF in both mice and humans [29].

## 5. Molecular Functions of SMARCB1

### 5.1. SMARCB1 Acting as a Tumor Suppressor via Regulation of p16^INK4a^

SMARCB1 has been shown to regulate the critical tumor suppressor, p16 (also known as p16^INK4a^) [115]. p16 is a cyclin-dependent kinase inhibitor that binds to CDK4/6 and prevents activation of the CDK4/6-cyclin D1 complex [116]. Active Cdk4/6-cyclin D1 is able to phosphorylate pRB, which releases E2F1 to promote gene expression profiles associated with S phase progression [117]. SMARCB1-deficient cells have reduced p16 expression, which ultimately leads to increased cellular proliferation due to unchecked S phase progression. Upon re-expression of SMARCB1, there is increased p16 expression levels, presumably due to increased BAF localization and remodeling activity at p16 regulatory regions [118]. This increase in p16 expression leads to cell cycle arrest at the G0/G1 phase due to increased Cdk4/6 inhibition.

### 5.2. SMARCB1 Inhibits MYC Target Activation

In addition to its role within the BAF complex, SMARCB1 has been shown to bind directly to MYC, a master regulator of genome-wide transcription that potentiates oncogenic transformation when overexpressed [56,119]. It was originally shown through a yeast two-hybrid screen that the Rpt1 domain of SMARCB1 binds to the basic helix-loop-helix (bHLH) and leucine zipper (Zip) of c-Myc [56]. The interaction between c-Myc and SMARCB1 was hypothesized to facilitate the expression of c-Myc targets, presumably through the recruitment of the BAF complex to target promoters. This hypothesis, however, conflicts with the established tumor suppressor role of SMARCB1. Moreover, subsequent studies have shown that SMARCB1 loss is associated with activation of MYC target genes [120,121,122].

More recent studies have opposed this idea of transactivation, instead proposing that SMARCB1 and MYC function antagonistically [123,124]. Biochemical and structural studies have shown that the presence of SMARCB1 reduces the DNA binding affinity of the MYC-MAX complex in vitro by inhibiting the E-box binding activity [124,125]. Genomic analysis revealed that the re-expression of SMARCB1 led to reduced binding of MYC genome-wide, particularly at promoters. Notably, this interaction was shown to be independent of the remodeling function of the BAF complex. These findings suggest that SMARCB1 has a separate tumor-suppressive activity outside of its regulation of chromatin accessibility. Within this model, SMARCB1 typically acts as an inhibitor of MYC DNA binding, which prevents MYC target gene expression in a wild-type setting. Yet, upon loss of SMARCB1, MYC can freely bind to its targets and promote oncogenesis. These findings ultimately point to MYC inhibition as a potential therapeutic target in patients with SMARCB1-deficient cancer.

### 5.3. Exportin 1 (XPO1)-Mediated Localization of SMARCB1

The BAF complex exerts its biological function only in the nucleus, so SMARCB1 requires nuclear localization to properly function as a tumor suppressor. While many cases of ATRT/MRT are characterized by a complete loss of SMARCB1 [126], a growing preponderance of ATRT cases appear to be caused by aberrant cytoplasmic localization of SMARCB1, particularly in the ATRT-TYR subgroup [127]. In fact, a recent study analyzing a series of 102 ATRT patient samples identified that 19% of the cases had cytoplasmic SMARCB1 localization [128].

A nuclear export signal (NES-residues 266–276) was found in the Rpt2 domain of SMARCB1, which is normally masked by the C-terminal domain (residues 319–385) [61]. Introduction of a truncated SMARCB1 lacking the C-terminal domain into an MRT cell line leads to the dramatic cytoplasmic localization of SMARCB1. The re-expression of this same truncated SMARCB1 does not induce the expected flat cell phenotype normally seen upon full-length SMARCB1 expression, indicating that the cytoplasmic localization of SMARCB1 eliminates its tumor suppressor function.

This cytoplasmic localization was found to be dependent upon exportin 1 (XPO1), which directly interacts with the NES sequence found in the Rpt2 domain of SMARCB1 causing cytoplasmic localization. Upon inhibition of XPO1 with leptomycin B, nuclear localization occurred in cells transduced with these truncated SMARCB1 mutants. These results were recently recapitulated with another nuclear export inhibitor, selinexor (KPT-330), a more clinically viable drug that was also shown to inhibit growth of these truncated SMARCB1 cells [128]. Further study is needed to determine the clinical efficacy of nuclear export inhibition in C-terminal truncated SMARCB1 rhabdoid tumors.

## 6. Advances in Molecular Subgrouping of ATRT

ATRTs primarily affect young infants or children, often with dismal prognosis [129,130,131]. Despite the genetic similarity between ATRT cases, they surprisingly present differing molecular and clinical features [132,133,134,135]. Due to the relatively limited number of ATRT cases, there have been challenges in the characterization of biological and molecular heterogeneity.

With each prior study using various sequencing platforms and subtyping parameters, an international effort has combined the results of numerous studies to reach a consensus on the molecular subtypes present in ATRT cases as well as their clinical features to facilitate future clinical studies and research (Table 1) [127]. Analysis of both transcriptomic and methylation datasets in this meta-analysis has identified three well-defined ATRT subgroups each with their own distinct clinical and molecular presentations: ATRT-TYR, ATRT-MYC, and ATRT-SHH [136].

The ATRT-TYR subgroup (34%) is named due to the overexpression of tyrosinase, an enzyme involved in the synthesis of melanin. Often, tyrosine immunohistochemistry can be used to diagnose ATRTs in this subgroup [137]. Global accessibility assays of ATRT-TYR samples suggest that this subgroup has more open chromatin than other subgroups, indicating that ATRT-TYR may originate from earlier developmental cell types [134]. This subgroup often presents with a whole or partial deletion in one SMARCB1 allele, with an inactivating point mutation in the other allele.

The ATRT-SHH subgroup (41%) is named due to the overexpression of both sonic hedgehog (SHH) and Notch members, both pathways known to be conserved key regulators of development [138,139]. Gene ontology analysis of this subtype suggests that these tumors often arise from neuronally differentiated cells, as compared to the other subgroups. This tumor subtype most often presents with compound point mutations in both SMARCB1 alleles. Additionally, this subtype has the highest rate of metastasis and lowest overall survival [136].

The ATRT-MYC subgroup (23%) is named due to the overexpression of the MYC oncogene, not to be confused with MYCN, a tissue-specific transcription factor typically found during early development and is found to be overexpressed in the ATRT-SHH subgroup and other pediatric solid tumors such as neuroblastoma. These tumors often present with a large homozygous deletion in both *SMARCB1* alleles, and rarely present point mutations within *SMARCB1*. This subtype often arises in slightly older children (3.4 years old) compared to the other subtypes.

A rarer fourth subgroup has been proposed, ATRT-SMARCA4 (~0.5–2%), which has biallelic SMARC4 inactivation yet retains wild-type SMARCB1 [140]. This subgroup of ATRTs is predominantly caused by germline mutations in SMARCA4, unlike the other subgroups, and mostly affects males.

Recent studies have suggested that these subgroup-specific epigenetic and transcriptomic profiles may be utilized to identify therapeutic vulnerabilities and provide targeted treatment that may improve current clinical outcomes [141]. Some of these novel molecular therapies for ATRT are discussed in the following section.

## 7. Therapeutic Vulnerabilities of SMARCB1-Deficient Cancers

There have been significant gains in our understanding of the molecular function of SMARCB1 in recent years. These novel insights have propelled the discovery of numerous potential therapeutic vulnerabilities of SMARCB1-deficient cancers. Several of these discoveries are based upon our increased understanding of how the epigenome is deregulated upon SMARCB1 loss. Other targets have been discovered through unbiased orthogonal screens that have identified targets that are cellular dependencies. This section outlines the recent advances in the therapeutic targeting of SMARCB1-deficient cancers (Figure 4).

### 7.1. Opposing Overactive PRC2 Repression by Inhibiting EZH2

Clinical benefits of PRC2 targeting, through inhibition of the catalytic EZH2 subunit, have been reported in preclinical data and clinical trials [103,142,143]. As previously described, the BAF complex opposes the repressive PRC2 complex to facilitate coordination of genome-wide gene expression [101]. Upon loss of SMARCB1, the BAF complex loses its ability to oppose PRC2 repression. This uncontrolled PRC2 repression leads to the widespread inhibition of BAF target genes. To oppose this uncontrolled repression by PRC2, inhibitors such as tazemetostat have been developed to inhibit the catalytic PRC2 subunit, EZH2 [143]. Tazemetostat has recently received FDA approval for the treatment of SMARCB1-deficient epithelioid sarcoma [144,145].

### 7.2. Targeting GBAF Dependency through BRD9 Degradation

Targeting the GBAF-complex-specific BRD9 or GLTSCR1 subunits has been proposed as a target for antitumoral potential [146]. GBAF can efficiently remodel at CTCF sites as well as promoters without SMARCB1. Genomic analysis indicates that the residual GBAF activity in SMARCB1-deficient cancers maintains the expression of retained BAF targets. In vitro assays indicate that rhabdoid tumors and synovial sarcoma are dependent on the residual remodeling activity of GBAF for proliferative maintenance. Furthermore, degradation or inhibition of the BRD9 or GLTSCR1 subunits selectively reduces cell viability in rhabdoid tumors [44,82,146].

### 7.3. Tyrosine Kinase Inhibition (PDGFRα/β and FGFR2)

The inhibition of overactive tyrosine kinases is an attractive target due to the numerous clinically approved compounds. Molecular profiling of rhabdoid tumors shows that PDGFRα/β and FGFR2 are coactivated, and that there is a dual blockade of these signals via the tyrosine kinase inhibitor (TKI), ponatinib [147]. A prior chemical inhibitor screen countered these findings, reporting that some ATRT cell lines were not susceptible to dual PDGFRα/β and FGFR2 inhibition [148]. These conflicting studies raise the possibility that the specific molecular subgrouping of ATRT samples may predict the response to kinase inhibition. Further studies are still needed to confirm if TKIs are a viable therapeutic target for SMARCB1-deficient cancers.

### 7.4. MYC Inhibition

MYC inhibition has been demonstrated to inhibit ATRT tumor growth in vivo and in vitro, particularly in the ATRT–MYC subgroup [124,149,150]. In vitro studies show that direct inhibition of MYC through the overexpression of OmoMYC seems to parallel the effects seen when SMARCB1 is reintroduced and reduces cell viability [124]. In vivo studies show that the treatment of orthotopic ATRT xenografts with the BET inhibitor, JQ1, mimics the effects observed when MYC is directly depleted and ultimately reduces tumor growth [150]. Based on these findings, MYC inhibition may be a viable therapeutic in rhabdoid tumor cases with high expression of MYC.

### 7.5. Immune Checkpoint Inhibition

Emerging evidence suggests that SMARCB1-deficient cancers may have anti-tumor immunogenicity and may be susceptible to immunotherapy [151,152,153]. In a study examining 30 pediatric patient samples with SMARCB1 loss, 47% of samples were positive for PD-L1 expression [154]. Clinical benefits of immune checkpoint inhibition in SMARCB1-deficient cancers are currently being studied in several clinical trials [155]. For this reason, further work is needed to understand how SMARCB1-deficient cancers respond to immunotherapy, especially when used in combination with other anti-tumor agents.

### 7.6. High-Throughput Screens to Identify Therapeutic Vulnerabilities

High-throughput screens are an efficient unbiased approach to identify putative drug targets in many different cancer types. In these screens, a library of drug compounds, shRNAs, and sgRNAs are used to perturb the function of a wide set of genes. By probing the abundance of the shRNA/sgRNAs over a time course, potential genetic vulnerabilities can be identified. Two recent studies have utilized these methods to identify putative therapeutic vulnerabilities in both rhabdoid tumors and renal medullary carcinoma. This section discusses the results of each of these studies.

#### 7.6.1. MDM2/4 Inhibition

Pre-clinical results indicate that the inhibition of MDM2/4 with idasanutlin or ATSP-7041 reduces rhabdoid tumor cell growth in vitro and increases survival in vivo [156]. This study utilized a high-throughput orthogonal screen with RNAi, CRISPR-Cas9-mediated knockout, and small molecules to identify potential drug targets in a set of 8–10 rhabdoid tumor cell lines. MDM2 and MDM4 were shown to be dependencies across many of the cell lines. Moreover, this dependency was shown to be more pronounced in rhabdoid tumors. More pre-clinical studies are needed to elucidate the clinical benefit of MDM2/MDM4 inhibition in SMARCB1-deficient cancers.

#### 7.6.2. Proteasomal Inhibition

A similar but focused orthogonal screen revealed that proteasomal inhibition reduces tumor cell growth of SMARCB1-deficient cancers in vitro and increases survival in vivo [15]. Gene targets involved in the ubiquitin-proteasome system were highly enriched for reducing cell growth. Treatment with the proteasome inhibitor, MLN2238 or ixazomib, reduced the function of the anaphase promoting complex (APC) E3 ligase subunit, UBE2C, leading to accumulation of cyclin B1. The inability of UBE2C to target cyclin B1 for degradation leads to G2/M cell cycle arrest. Another group had similar findings, showing that proteasomal inhibition paired with genetic suppression of autophagy led to durable responses in a faithful rhabdoid tumor mouse model [157]. These effects of proteasomal inhibition were similarly validated within the context of rhabdoid tumors, yet the synergistic effects when paired with autophagy inhibition were not observed in the context of ATRT [158]. Further studies are still needed to elucidate the clinical benefits of proteasomal inhibition in SMARCB1-deficient cancers.

## 8. Conclusions

Since the initial discovery of SMARCB1 as a tumor suppressor in the 1990s, our understanding of SMARCB1 and its molecular interactions has grown tremendously. Despite these major advancements, an effective treatment for patients with SMARCB1 deficient cancers has remained elusive. With the advent of widespread sequencing technologies in cancer diagnosis, the number of diseases and subsequent cases that involve SMARCB1 alterations has grown. Therefore, it is important that we utilize these gains in our understanding to push forward potential therapeutic treatments.

In recent years, there has been a shift toward a more personalized molecular subtyping of each SMARCB1-deficient cancer, such as those four subgroups defined for ATRT. This trend is likely to continue as more SMARCB1-deficient tumor samples are sequenced and analyzed, hopefully leading to more targeted therapies that are suited for each molecular subtype. Within each subtype, there may also be targeted treatments dependent on specific SMARCB1 alterations, such as those cases with the cytoplasmic localization of SMARCB1. These novel molecular insights discussed in this review have set the stage for future clinical trials that may soon lead to effective treatments for SMARCB1-deficient tumors.

## Figures and Tables

**Figure 1 cancers-14-03645-f001:**
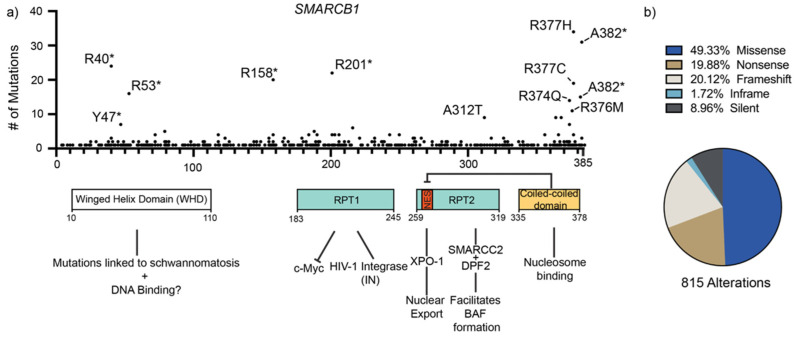
Cancer-associated mutations and molecular interactions of SMARCB1. Data obtained from COSMIC (v95) reveal a clustering of mutations in the C-terminal coiled-coiled domain as well as a high preponderance of truncation mutations on the N-terminal end. (**a**) Frequency of cancer associated mutations in *SMARCB1*. Nonsense mutations and frameshift mutations are denoted by (*). The four known functional domains of SMARCB1 and their reported molecular interactions are shown. (**b**) The proportion of each mutation type within the COSMIC dataset. In-frame and frameshift mutations include both deletions and insertions.

**Figure 2 cancers-14-03645-f002:**
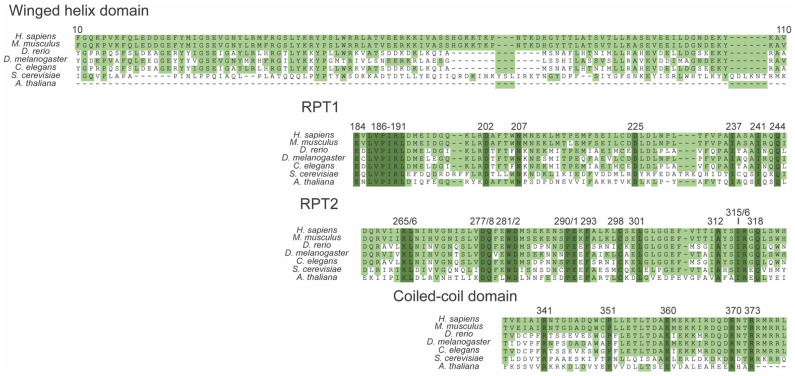
Conservation of SMARCB1 across seven eukaryotic species: H. sapiens-NP_003064.2, M. musculus-BAB12427.1, D. rerio-NP_001007297.1, C. elegans-NP_001369845.1, S. cerevisiae-ONH79494.1, and A. thaliana-NP_001189918. Light green boxes represent amino acid residues that match the H. sapiens SMARCB1 sequence. Dark green residues represent amino acid residues that are conserved in all seven eukaryotic species used in this analysis.

**Figure 3 cancers-14-03645-f003:**
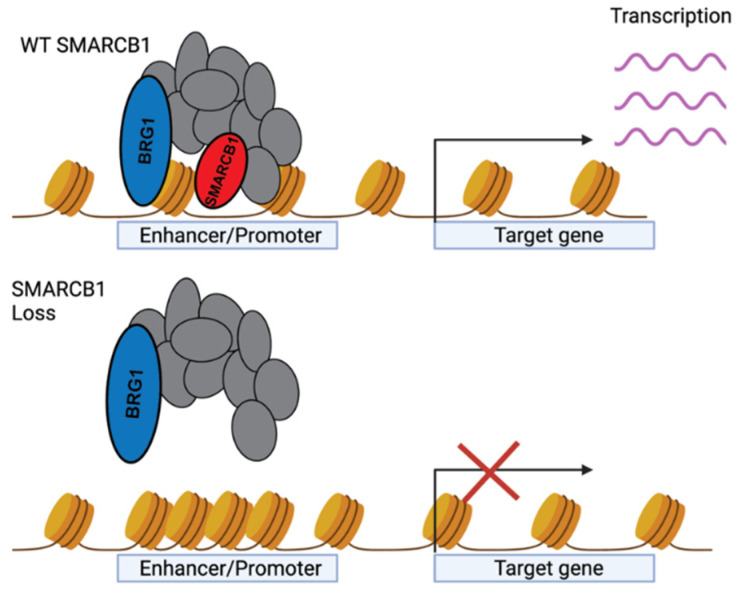
Role of SMARCB1 within the BAF complex in regulating gene expression.

**Figure 4 cancers-14-03645-f004:**
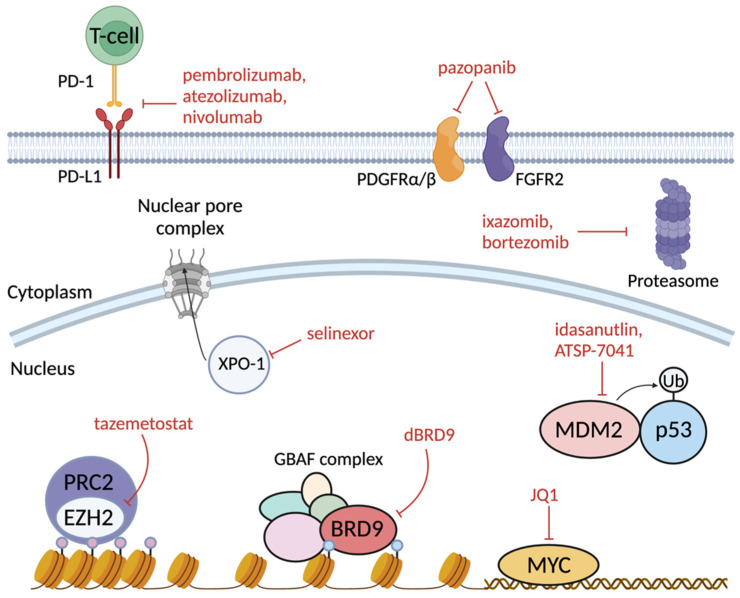
Therapeutic vulnerabilities discovered in SMARCB1-deficient cancers.

**Table 1 cancers-14-03645-t001:** ATRT subtypes and their respective clinical and molecular characteristics.

KERRYPNX	ATRT-MYC	ATRT-SHH	ATRT-TYR	Reference(s)
Median age at diagnosis (years)	3.4	1.4	1.5	[136]
% with RTPS	0%	~36%	~20%	[136]
% Metastatic	~30%	~46%	~10%	[136]
Predominant CNV at SMARCB1 locus	Focal loss (50%)	Focal loss (50%)	Focal (5%)	[133,136]
	Broad loss (7%)	Broad loss (7%)	Broad loss (62%)	
	Small loss (29%)	Small loss (29%)	Small loss (20%)	
	None (14%)	None (14%)	None (24%)	
Site of tumor	Infratentorial (22%)	Infratentorial (30%)	Infratentorial (80%)	[133,136]
	Supratentorial (64%)	Supratentorial (70%)	Supratentorial (20%)	
	Spine (14%)			
Molecular characterization	Overexpression of the *MYC* oncogene and *HOX* cluster	Overexpression of sonic hedgehog and notch members	Overexpression of tyrinosinase and melanosomal gene	[127,133]
Sex	Male (54%)	Male (47%)	Male (62%)	[136]
	Female (46%)	Female (53%)	Female (38%)	
5-year Overall Survival	16.7 ± 10.8%	15 ± 9.8%	58.8 ± 11.9%	[136]
Methylation Status	Hypomethylated	Hypermethylated	Hypermethylated	[133]

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
