# Peer review of "SMARCB1-Deficient Cancers: Novel Molecular Insights and Therapeutic Vulnerabilities"

_cancers, 2022, doi:10.3390/cancers14153645_

Round 1

Reviewer 1 Report

The authors performed an exhaustive review of the literature, from the initial discoveries regarding the SMARCB1 gene to the recent findings of the involvement of this gene in several human cancers.

The manuscript is a useful and comprehensive work, able to give a sufficiently complete picture of the structure, the molecular functions of SMARCB1 and the potential clinical implications in the treatment of SMARCB1-deficient cancers

Author Response

The authors performed an exhaustive review of the literature, from the initial discoveries regarding the SMARCB1 gene to the recent findings of the involvement of this gene in several human cancers.

The manuscript is a useful and comprehensive work, able to give a sufficiently complete picture of the structure, the molecular functions of SMARCB1 and the potential clinical implications in the treatment of SMARCB1-deficient cancers.

We thank the reviewer for their interest in our manuscript and these positive remarks.

Reviewer 2 Report

 In this review, the authors aimed to outline critical mechanisms by which loss of SMARCB1 may lead to tumor formation and cover how these mechanisms have been used to design targeted therapy. The paper is interesting since compelled the literature about the SAMRCB1 deficient Cancers, is well written and organized. I have just some minor suggestions to the authors:

1.     I have a curiosity about the hereditary origin of the SMARCB1-deficient cancers: there is some information about that issue? If yes, I suggest including in the paper;

2.     In line 45 the sentence “This review will use the Human genome…” has no continuation on the following page;

3.     Subchapter 2.2. is misplaced, in my opinion, the order should be 2.1, 2.3, and in the end 2.2;

4.     Figure 2, page 5, is misnamed as Figure 1, also this figure is cited after the Figure on page 6. It's preferable that the figure appears after de citation and not before;

5.     Being a revision paper, I missed some recent references that should be included and discussed, as for example:

- Ngo C, Postel-Vinay S. Immunotherapy for SMARCB1-Deficient Sarcomas: Current Evidence and Future Developments. Biomedicines. 2022 Mar 11;10(3):650. doi: 10.3390/biomedicines10030650. PMID: 35327458; PMCID: PMC8945563.

- Lee VH, Tsang RK, Lo AWI, Chan SY, Chung JC, Tong CC, Leung TW, Kwong DL. SMARCB1 (INI-1)-Deficient Sinonasal Carcinoma: A Systematic Review and Pooled Analysis of Treatment Outcomes. Cancers (Basel). 2022 Jul 5;14(13):3285. doi: 10.3390/cancers14133285. PMID: 35805058; PMCID: PMC9265388.

- Zhang C, Li H. Molecular targeted therapies for pediatric atypical teratoid/rhabdoid tumors. Pediatr Investig. 2022 May 23;6(2):111-122. doi: 10.1002/ped4.12325. PMID: 35774526; PMCID: PMC9218972.

- Wang N, Qin Y, Du F, Wang X, Song C. Prevalence of SWI/SNF genomic alterations in cancer and association with the response to immune checkpoint inhibitors: A systematic review and meta-analysis. Gene. 2022 Aug 5;834:146638. doi: 10.1016/j.gene.2022.146638. Epub 2022 Jun 6. PMID: 35680019.

 (Among others recently published on pubmed)

Author Response

In this review, the authors aimed to outline critical mechanisms by which loss of SMARCB1 may lead to tumor formation and cover how these mechanisms have been used to design targeted therapy. The paper is interesting since compelled the literature about the SAMRCB1 deficient Cancers, is well written and organized.

We thank the reviewer for their interest in our manuscript and these positive remarks.

I have just some minor suggestions to the authors:

  1. I have a curiosity about the hereditary origin of the SMARCB1-deficient cancers: there is some information about that issue? If yes, I suggest including in the paper;

A section on the hereditary origin of SMARCB1 deficient cancers, known as RTPS, has been added to section 2.2.

In lines 132-138 of the revised manuscript, we have added:

“Familial cases of rhabdoid tumor can occur when an individual has a condition known as rhabdoid tumor predisposition syndrome (RTPS). In RTPS, a germline vari-ant arises in either SMARCB1 (known as RTPS1) or the ATPase subunit found in the BAF complex, SMARCA4 (known as RTPS2) [32, 33]. Having a germline mutation in one allele of either SMARCB1 or SMARCA4 increases the risk of developing rhabdoid tumors in children and schwannomas in adults because only one functional copy is present in each cell.”

  1. In line 45 the sentence “This review will use the Human genome…” has no continuation on the following page;

We thank the reviewer for identifying this and apologize for this formatting error. This has been corrected, and there should be continuity between pages.

The text has been corrected and in the revised manuscript is now lines 47-48. Specifically we have correct the text to state: “This review will use the Human Genome Organization derived name of SMARCB1”

  1. Subchapter 2.2. is misplaced, in my opinion, the order should be 2.1, 2.3, and in the end 2.2;

We thank the reviewer for this suggestion and we agree that the flow of the manuscript is improved by switching the order to sections 2.1, 2.3 and 2.2.

  1. Figure 2, page 5, is misnamed as Figure 1, also this figure is cited after the Figure on page 6. It's preferable that the figure appears after de citation and not before;

We thank the reviewer for identifying this and apologize for this error. The figure has since been renamed to Figure 2, and we have also formatted the manuscript so that the citation occurs before the figure.

  1. Being a [review] paper, I missed some recent references that should be included and discussed, as for example:

- Ngo C, Postel-Vinay S. Immunotherapy for SMARCB1-Deficient Sarcomas: Current Evidence and Future Developments. Biomedicines. 2022 Mar 11;10(3):650. doi: 10.3390/biomedicines10030650. PMID: 35327458; PMCID: PMC8945563.

- Lee VH, Tsang RK, Lo AWI, Chan SY, Chung JC, Tong CC, Leung TW, Kwong DL. SMARCB1 (INI-1)-Deficient Sinonasal Carcinoma: A Systematic Review and Pooled Analysis of Treatment Outcomes. Cancers (Basel). 2022 Jul 5;14(13):3285. doi: 10.3390/cancers14133285. PMID: 35805058; PMCID: PMC9265388.

- Zhang C, Li H. Molecular targeted therapies for pediatric atypical teratoid/rhabdoid tumors. Pediatr Investig. 2022 May 23;6(2):111-122. doi: 10.1002/ped4.12325. PMID: 35774526; PMCID: PMC9218972.

- Wang N, Qin Y, Du F, Wang X, Song C. Prevalence of SWI/SNF genomic alterations in cancer and association with the response to immune checkpoint inhibitors: A systematic review and meta-analysis. Gene. 2022 Aug 5;834:146638. doi: 10.1016/j.gene.2022.146638. Epub 2022 Jun 6. PMID: 35680019.

 (Among others recently published on pubmed)

We thank you for these new manuscripts to include in this review.  We have since added each of the recommended references into our manuscript, and have added a section (7.5) describing immunotherapy in SMARCB1-deficient cancers.